# P1G10, the Proteolytic Fraction from *Vasconcellea cundinamarcensis*, Stimulates Tissue Repair after Acute Exposure to Ultraviolet B Radiation

**DOI:** 10.3390/ijms20184373

**Published:** 2019-09-06

**Authors:** Kátia M. Freitas, Ana C. Araújo e Silva, Emerson S. Veloso, Ênio Ferreira, Lucíola S. Barcelos, Marcelo V. Caliari, Carlos E. Salas, Miriam T. P. Lopes

**Affiliations:** 1Departamento de Farmacologia, Instituto de Ciências Biológicas, Universidade Federal de Minas Gerais, Av Antônio Carlos 6627, Belo Horizonte 31270-901, MG, Brazil (K.M.F.) (A.C.A.e.S.) (M.T.P.L.); 2Faculdade de Medicina do Mucuri, Universidade Federal dos Vales do Jequitinhonha e Mucuri, Rua do Cruzeiro, nº 01, Bairro Jardim São Paulo, Teófilo Otoni 39803-371, MG, Brazil; 3Departamento de Patologia, Instituto de Ciências Biológicas, Universidade Federal de Minas Gerais, Av Antônio Carlos 6627, Belo Horizonte 31270-901, MG, Brazil (E.S.V.) (Ê.F.) (M.V.C.); 4Departamento Fisiologia e Biofísica, Instituto de Ciências Biológicas, Universidade Federal de Minas Gerais, Av Antônio Carlos 6627, Belo Horizonte 31270-901, MG, Brazil; 5Departamento de Bioquímica e Imunologia, Instituto de Ciências Biológicas, Universidade Federal de Minas Gerais, Av Antônio Carlos 6627, Belo Horizonte 31270-901, MG, Brazil

**Keywords:** antioxidant, anti-inflammatory, *C. candamarcensis*, erythema, proteolytic, papain, ultraviolet, UVB, healing

## Abstract

Background: P1G10 is a cysteine proteolytic fraction from *Vasconcellea cundinamarcensis* latex, obtained by chromatographic separation on Sephadex-G10 and ultrafiltration. This fraction enhances healing in different models of skin lesions, and displays a protective/healing effect against gastric ulcers, where it was suggested an antioxidant role. Methods: We evaluated here the effect of topical treatment with P1G10, in mice lesions induced by UVB. Results: After single exposure to 2.4 J cm^−2^ UVB, P1G10 reduced erythema, increased cellularity of hypodermis, enhanced MPO activity and IL1β, and inhibited COX_2_ levels. These results point to an anti-inflammatory effect by P1G10. This fraction displayed antioxidant activity by reversing the depletion of glutathione (GSH), glutathione peroxidase (GSH-Px), superoxide dismutase (SOD) and reducing the catalase activity increased by UVB. These changes may be related to a reduction in MDA observed in groups treated with P1G10. P1G10 also inhibited MMP-9, caspase-3 and pkat while increasing p53 levels.

## 1. Introduction

The skin is an organ acting as a selective barrier between the body and the external environment. It is constantly exposed to a pro-oxidant environment, including ultraviolet light [1]. Solar UV light spans radiation between 200 and 400 nm. While UVC (200–280 nm) is mostly filtered by the stratospheric ozone layer, UVB (280–320 nm) and UVA (320–400 nm) reach the earth surface and biomass. Solar UVB radiation is absorbed by the skin and is responsible for, cutaneous injury [2], inflammation (erythema) [3], immunosuppression [4,5], photoaging, melanoma- and non-melanoma skin cancer [6,7], and eye damage [8].

The mechanism underlying UV lesions involves damage through photons targeting DNA, proteins, and smaller molecules. The activated derivatives alter their biological properties or act as inducers of chemical reactions involving reactive free radicals (ROS). 

An initial inflammatory response involves cutaneous vasodilation resulting in erythema and includes epidermal hyperplasia, enlargement of stratum corneum, increased vascularization, and melanogenesis [8,9]. Mediators involved in erythema include prostaglandins (PGE2), nitric oxide, pyrimidine dimer formation in DNA (chromophore), and inflammatory cytokines [10]. Damaged cells undergo apoptosis and are phagocytized by neighboring keratinocytes and infiltrating macrophages. Infiltrating leukocytes and resident skin cells release ROS that induce synthesis of pro-inflammatory cytokines such as TNF-α, IL-6, IL-1β, IL-8, and COX_2_ and iNOS, triggering further cellular and tissue damage [11,12].

Changes in interleukins IL-10, IL-4 and PGE2 affecting the immune response are also observed [10,13]. The impaired cellular immunity after UV exposure affects the host response to a variety of bacterial and viral challenges [10].

No significant decrease on earth UVB radiation resulting from recovery of ozone depletion is yet observed [14]. Thus, only by the end of this century (2090–2100), due to ozone recovery and increasing greenhouse effects, a decrease in UV irradiance is expected [15]. To face this challenge several options are developed to cope with the hazardous effect of UV radiation. A current trend favors use natural products as photoprotective agents [16,17]. UV-filtering molecules generally display conjugated π electrons found in linear or aromatic molecules [18]. These compounds help to reduce skin irradiation, decrease inflammation, DNA damage, and oxidative stress and may favor cellular pathways to protect from the UV effect [19]. Instead, our group focuses on plant proteolytic enzymes as source of phyto-pharmacophores to treat skin lesions. By using several wounding models, we were able to demonstrate the efficacy of a fraction designated as P1G10. This fraction shows anti-inflammatory and antioxidant potential [20,21]. In this study, we evaluated the effect of this fraction on skin lesions induced by UVB radiation in an animal model. The results showed that P1G10 downregulates the mediators associated to erythema and decreases the inflammatory response associated to UV stress.

## 2. Results

### 2.1. UVB Radiation-Induced Changes

The evolution of cutaneous macro and microscopic lesions induced with 2.4 J cm^−2^ UVB, were observed in mice at 6, 12, 24, and 48 h following exposure. Erythema peaked 12 h after irradiation (Figure 1A).

Photomicrographs of HE stained skin sections showed that at zero-time, intact skin displayed mature collagen fibers and a moderate amount of fibroblast. After 6 h of irradiation, an increase in epidermal cell volume and apoptosis was observed. Then, at later intervals, enucleated cells and increased number of eosinophils and degraded collagen fibers (Figure 1A) were observed. These latter changes declined at 48 h post irradiation. At 24 h, the skin of irradiated samples was thicker (752.1 ± 73.56 µm, *p* < 0.01) compared to the basal group (445.0 ± 18.21 µm), followed by a reduction at 48 h (Figure 1B). The intensity of erythema also decreased 48 h after irradiation (Figure 1A).

The profile of redox markers at different intervals after UVB-radiation was investigated. The SOD activity peaked (3-fold) at 12 h, then declined to basal at 24 and 48 h. Interestingly, CAT displayed two maxima, one, 6 h post-irradiation reaching 180% (*p* < 0.05), followed by a decline to basal level at 12 h and ensued by a second peak 220% at 24 h (*p* < 0.01) that remained elevated (210%) at 48 h (*p* < 0.05) (Figure 2A). In contrast, GSH shows a significant depletion 6 h after UVB-irradiation (*p* < 0.001), followed by gradual rise without attaining normal level (Figure 2B). The effect of UVB on MPO, the pro-inflammatory marker was also evaluated in skin samples. In supernatants from irradiated samples, MPO activity shows a gradual increase after irradiation, reaching 300% at 48 h post irradiation (*p* < 0.001, basal), as shown on Figure 2C.

### 2.2. Effect of P1G10 on UVB Irradiated Mice

The effect of topical administration of P1G10 (0.1% or 1.0%) on UVB-irradiated skin at 24 h was chosen for analysis, as many of the changes induced by UV were maximized during this period. In these conditions (Figure 3A), in untreated irradiated- and vehicle treated irradiated mice, increased erythema was observed (+++) confirming the initial observation (Figure 1A). Meanwhile, P1G10 at both concentrations reduced the intensity of erythema. Correspondingly, in H&E stained tissue sections, there was a 43% (*p* < 0.01) decrease in skin thickness following 0.1% P1G10 at 24 h, and 35% at 1% P1G10, (*p* < 0.05) compared to the (control) vehicle group (Figure 3B).

#### 2.2.1. P1G10 Modulates UVB-Induced Antioxidant Responses

To investigate the ability of P1G10 to quench the surge of ROS induced by UV, we evaluated the activities of SOD, CAT, GSH-Px and GSH content after 24 h UVB-exposure. The depletion of SOD activity induced by UVB after 24 h observed in the saline and control group was reversed by 0.1% P1G10, but the effect vanishes at 1% P1G10 (Figure 4A). Meanwhile, the increase in CAT activity after UVB was reversed by 0.1% and 1% P1G10, as in the basal group (Figure 4A).

Analysis of GSH and GSH-Px (Figure 4B,C), shows that UVB radiation decreased the GSH level by 31% and GSH-Px by 50% at 24 h. P1G10 at 0.1% and 1% prevented the decrease in GSH and 1% P1G10 restored GSH-Px activity, like in the basal group.

#### 2.2.2. P1G10 Reduced Infiltration and Activation of Inflammatory Cells

A morphometric analysis showed a 30% reduction in inflammatory infiltrate in the group treated with 1.0% P1G10 relative to the vehicle, (*p* < 0.01) (Figure 5A). Also, P1G10 reduced the infiltration of neutrophils by 50%, assessed by MPO activity (Figure 5B), in the group that received 1.0% (*p* < 0.05) compared to the vehicle group.

#### 2.2.3. P1G10 Modulates Inflammatory Mediators Induced by UVB

We measured the level of inflammatory mediators after 24 h (Figure 6A), to establish a link with the morphological changes observed above (Figure 1). The depletion of TGF-β and the increase in IL8 induced by UVB were not affected in presence of P1G10. Meanwhile, the increase in IL1β upon irradiation was reversed in the group treated with 1% P1G10. Finally, IL10 was not altered by UVB, but in presence of 1% P1G10, there was a 75% increase compared to basal value. As expected, the level of COX_2_ (Figure 6B), an inflammatory marker, was upregulated in irradiated skin at 24 h and, 0.1% and 1% P1G10 strongly inhibited COX_2_, like in the Basal group (*p* < 0.01).

#### 2.2.4. P1G10 Prevents UVB-Induced Collagen Degradation

Figure 7A shows representative Gomori trichrome stained sections stressing collagen fibers, their organization and preservation after irradiation. UVB-irradiated dermis showed a significant decrease in abundance, density, integrity, and organization of collagen fibers compared to unirradiated skin. Meanwhile, treatment with P1G10 (0.1% or 1%) inhibited the irradiation-induced effects. In the presence of P1G10, collagen fibers show improved alignment and better staining (2-fold, *p* < 0.05), at both concentrations (Figure 7B).

We also analyzed by zymography the activity of MMP-9 in skin samples, 24 h after UVB irradiation in the presence or absence of P1G10 (Figure 7B). Gel densitometry in irradiated tissue shows a 2-fold increase in MMP-9 activity (*p* < 0.001) compared to basal level. The presence of 0.1% or 1.0% P1G10 reversed the MMP-9 increase (*p* < 0.05), becoming like the basal group.

#### 2.2.5. P1G10 Reduces Oxidative Damage

The antioxidant capacity of P1G10 (6.2–100.0 μg mL^−1^) was evaluated by DPPH radical scavenging assay, using ascorbic acid as positive control (100% inhibition). The concentration of P1G10 that was able to sequester 50% of free radicals (EC_50_) in solution was estimated by linear regression to be 20.35 ± 0.66 μg mL^−1^ (Figure 8A).

The generation of ROS in skin samples was determined by malondialdehyde (MDA) and dichlorofluorescin diacetate (DCFH-DA) used as markers for photo-oxidative damage. After UVB-exposure, we observed an 80% increase in MDA level (Figure 8B) in the control and saline groups, compared to basal level (*p* < 0.01). Application of 0.1% P1G10 post-irradiation moderately depleted MDA, but, in the 1.0% P1G10 group, MDA decreased by 48% (*p* < 0.01), compared to the vehicle control. We also analyzed ROS, by via oxidized dichlorofluorescein (DCF) (Figure 8C). In UVB irradiated skin, there was a 3-fold increase compared to the non-irradiated control (*p* < 0.01). Meanwhile, in UVB-induced groups treated with 0.1% or 1.0 % P1G10, ROS production was inhibited by 50% and 60%, respectively, both compared to the irradiated control. These ROS values in P1G10 treated samples became like the Basal level (Figure 8C).

#### 2.2.6. P1G10 Affects Levels of ROS-Related Signaling Proteins

Western blot analysis of pAkt, p53, and caspase-3 revealed changes after UVB-irradiation and exposure to P1G10 (Figure 9A). We observed in UVB-treated groups a 4-fold increase in pAkt (*p* < 0.05) and 8-fold increase of caspase-3 (*p* < 0.001), relative to the basal non irradiated group. The increase in pAkt was fully reversed at 0.1% and became lower than basal at 1% P1G10. A similar trend is observed for caspase-3, but the decline caused by P1G10, although significant, did not attain basal level (Figure 9A). In contrast, UVB-irradiation had no effect on p53, but 1.0% P1G10 induced a five-fold increase in this protein, (*p* < 0.01) (Figure 9A). Histochemical analysis of sections illustrates the increase in caspase-3 after UVB irradiation and its suppression in presence of 0.1% and 1% P1G10 (Figure 9B).

## 3. Discussion

To validate the effect of P1G10 on UV irradiated skin of hairless mice, we carried initial experiments to assess between 6 and 48 h, changes in selected agents associated to redox status and the macro- and microscopic effects induced by radiation (Figure 1). Erythema which is induced by vasodilation of blood vessels [22], had its maximum at 12 h after irradiation. A biphasic pattern of erythema formation described before in other animal models was missing in hairless mice, confirming an early study [23]. A significant increase in skin thickness amounting to 69% was also observed at 24 h post-UV which declined at 48 h (Figure 1B).

The status of antioxidant enzymes after irradiation showed a single SOD peak at 12 h and a bimodal response of CAT with two maxima at 6 h and 24 h post UV (Figure 2A). Pence and Naylor [24] found that SOD and CAT in SKH:hr1 mice (hairless) were inhibited 12 h after 90 mJ cm^−2^ UVB treatment and continued to be depressed up to 72 h post-UVB. In their study the authors report strong fluctuations in basal levels which they attributed to circadian variations of these enzymes. On the other hand, Iizawa et al. [25] reported an initial decrease in CAT and SOD, and a surge at 24 h in C57 BL/6 mice skin after a single UV dose (200 mJ cm^−2^). We find that GSH, crucial for removal of ROS species O_2_^−^ and derivatives [26], was depressed at 6 h, followed by gradual recovery without attaining basal level 48 h post irradiation (Figure 2B). Instead, Connor and Wheeler [27] reported transient fluctuations in epidermal GSH after UV radiation in hairless mice. The activity of MPO, a marker of neutrophil activation, increased between 0 and 48 h in the UV group in agreement with an early report [28] (Figure 2C). MPO is recognized for catalyzing reactions related to ROS production, and for limiting the magnitude of the immune response [29]. As seen here, the comparison of these variations in activities must consider the animal strain, radiation intensity, and as mentioned earlier, physiological changes influencing the model. The variations in levels of examined agents compared to the available data justified the initial analysis of these parameters in our model.

The rationale for analyzing the topical effect of P1G10 in UV lesions is based on reports showing that P1G10 accelerates wound healing and downregulates inflammatory symptoms within a shorter interval [30,31]. Figure 3A,B summarizes the changes in erythema and skin thickness induced by P1G10 in irradiated skin at 24 h.

Erythema resulting from capillary vasodilatation is a classical symptom of inflammation [10], and its reduction in P1G10 groups is attributed to its anti-inflammatory action. Photoaging induced by UV, and expressed as degradation of ECM proteins including collagen I, is mediated by MMPs [32]. The increase in MMP-9 observed upon UV stress was suppressed by P1G10 at both concentrations and provides support for the decrease in skin thickness (Figure 3B) and the increase in collagen observed in treated groups (Figure 7).

Also, the hyperplasia seen here in UV treated skin and reported earlier [33,34], was reversed by P1G10 (Figure 5A), supporting the anti-inflammatory effect. Coincidentally, green tea induced similar thinning of epidermal thickness in UV-irradiated mice and this was attributed to its antioxidant–anti-inflammatory effect [34].

The anti-inflammatory effect of P1G10, which limits cell damage by decreasing the amount of toxic (ROS) species, was supported by the decrease in MPO activity at 24 h (Figure 5B), whereas in the untreated irradiated group, MPO activity did not level off at 48 h post-UV (Figure 2C),

In this model, the antioxidant activity in SOD and CAT showed opposite changes after UV, and P1G10 drove these two enzyme levels back to basal, implying a reduction of the oxidative stress. Depletion of GSH induced by UV was also abolished in P1G10 groups, beyond basal level, creating a surplus of the reducing agent. These changes provide additional support for the UV antioxidant/anti-inflammatory action of P1G10.

We investigated the effect of UV on the activity of glutathione peroxidase (GSH-Px), the enzyme that removes H_2_O_2_ by oxidation of GSH. Figure 4C demonstrates that the decline of GSH-Px activity by UV was reversed following P1G10 treatment, contributing to removal of ROS species. It is reported that the decline in GSH-Px activity triggered by UV was restored and peaked at 24 h [25], while we observed a 50% deficit of this activity at the end of this interval. It has been suggested that the efficacy for GSH-Px removal of ROS becomes relevant when concentration of peroxides is high [4]; this would be the case in the acute UV model applied here. MDA, the byproduct of lipid peroxidation, displayed a 4-fold increase in the UV groups, as described in UV-irradiated human skin [35]. In the group treated with 1% P1G10, there was a significant drop in MDA, suggesting a reduction in lipid peroxidation due to a decreased production of ROS, thus supporting the antioxidant activity (Figure 8B).

As described so far, the anti-inflammatory role of P1G10 was caused at least in part by a reduction of ROS species, thus, we investigated the possible antioxidant power in P1G10. The ROS scavenging activity using the DPPH assay supported the ROS scavenging ability of P1G10 in a concentration dependent manner (Figure 8A). In a subsequent DCF assay to detect ROS activity in tissue samples, a surge in ROS was confirmed in UV groups and also the quenching effect on ROS triggered by P1G10. The explanation for the reversal in UV deleterious effects induced by P1G10 is the enhanced ROS scavenging effect seen in the treated groups. We do not have prior evidence for a ROS scavenging activity in P1G10, but papain, the proteolytic fraction from *C. papaya* which is equivalent to P1G10 in *V. cundinamarcensis* (https://www.drugbank.ca/drugs/DB11193), displays strong antioxidant activity and inhibits lipid peroxidation [36]. We remind that the anti-inflammatory and the accelerated healing effects attributed to P1G10 rest on proteolytic activity, as the iodoacetamide inhibited fraction loses its efficacy [21,31].

We also analyzed changes in COX_2,_ the limiting enzyme during synthesis of PGs, induced after UV-irradiation [37]. While COX_2_ was stimulated at 24 h post-UV, in P1G10 treated groups the enzyme was downregulated to Basal, preventing the formation of inflammatory mediators. In previous experiments, our group showed that induced gastric lesions treated with P1G10 had reduced expression of COX_2_ as well [20]. COX_2_ is targeted to treat diverse inflammatory symptoms, and several synthetic drugs [38] and plant metabolites have shown promise as inhibitors of COX_2_ [37]. A similar antioxidant property has been described in the proteolytic fraction (bromelain) from *Ananas comosus*, which downregulates the expression of COX_2_ and PGE-2 in murine and human cells [39]. On the other hand, we cannot ignore that papain has been described as an airway allergen that triggers an inflammatory reaction involving a Th2 response [40]. Whether this is a specific attribute of papain or results from individual genetic susceptibility to this allergen has not been established, but there is no evidence that bromelain or P1G10 from *V. cundinamarcensis* share the same allergenic feature.

To verify a role of cytokines in connection with the anti-inflammatory response, we analyzed TGF-β, IL1β, IL8, and IL10 after treatment with P1G10. While P1G10 had no effect on the depletion in TGFβ and IL8 induced by UV, a drop of the proinflammatory IL1β was observed with 1% P1G10, supporting the anti-inflammatory action assigned to P1G10. Meanwhile, IL10, an immunoregulator that plays a key role reducing immune responses and preventing excessive inflammation [41], was induced beyond basal at 1% P1G10 (Figure 6A). The meaning of this increase in IL10 is unknown and requires further analysis.

To gain insight into possible pathways affected in groups treated with P1G10, we assessed the levels of akt, pakt, p53, and capase-3. A strong decrease in pakt and increase in p53 were observed in the group receiving 1% P1G10, inducing a cell-cycle arrest. An intense crosstalk between p53 and pakt/mTOR was reported following UV stress to control the cellular fate through; cell-cycle arrest, cell death, autophagy, and senescence [42]. However, caspase-3 decreased in P1G10 treated groups, arguing for a reduction in apoptosis. We have observed a similar decrease in the apoptotic index in a wound rodent model treated with P1G10 [31]. We propose that cell-cycle arrest, autophagy, and senescence are privileged by P1G10 in detriment of apoptosis.

Many antioxidant/anti-inflammatory small molecules from plants have been reported in the literature, but only a few of them are proteins, except for endogenous proteins directly involved in detoxification of ROS. An exception is the *apelin* human peptide that inhibits vasodilation of blood vessels induced by UVB exposure [22]. In this study we showed that a group of proteases involved in plant protection play a novel role ameliorating the inflammatory symptoms prompted by UV stress in animals. The data confirm that P1G10 acts as healing enhancer in diverse wound scenarios.

The possibility of modulating the anti-inflammatory action by incorporating an active agent into liposomes is being established [43,44] and there are reports [45] describing the production of phospholipid vesicles containing bromelain or papain without impairing the proteolytic activity, thus we envision future studies to produce nanoparticles containing P1G10 for treatment of inflammatory symptoms.

## 4. Materials and Methods

### 4.1. Chemicals

EDTA, sodium acetate, sodium hydroxide, dimethylsulfoxide, absolute ethanol, potassium monophosphate, and Triton X-100 were from Merck, Darmstadt, Germany. Milli-Q-water was purified through Millipore, System Bedford, MA, USA. All the reagents used in formulations were for analysis Galena (Campinas, SP, Brazil) or Clariant (Sao Paulo, SP, Brazil). Sephadex G-10 was from GE Healthcare; Coomassie G-250 was from Bio-Rad Laboratories, Hercules, CA, USA; TdT-Fragel DNA-Fragmentation Detection Kit was from Calbiochem, San Diego, CA, USA).

### 4.2. Animals

Male *hairless* (HRS/J) mice (8–10 weeks old) whose matrices were provided by Universidade de São Paulo-Campus Ribeirão Preto were maintained in the animal house facility from the Laboratório de Substâncias Antitumorais (LSAT), UFMG. The experiments were carried out in accordance to internationally accepted guidelines on animal handling and were approved by the institutional Ethics Committee on Animal Experimentation, CETEA-Protocol 174/2010. The animals were housed in individual cages with controlled temperature, humidity and light–dark cycle, with unrestrained access to food and water.

### 4.3. Production of P1G10

Latex was obtained from unripe fruits of *Vasconcellea cundinamarcensis* females from Valle del Elqui, Chile. A voucher specimen of the plant was deposited at the herbarium of the Universidad de La Serena, Chile, with No 15063. Latex was collected by incisions onto the surface of unripe fruits with a steel blade. Following collection, latex was stored in the dark, at −20 °C until lyophilized. The isolation of P1G10 was described previously [21]. Briefly, freeze-dried latex was dissolved in buffer containing 25 mM cysteine, 5 mM DTT, and 10 mM EDTA pH 5.0 in 1 M sodium acetate solution. After low speed centrifugation and filtration (Whatman #1, Wilmington, MA, USA), the clear filtrate was chromatographed through Sephadex G-10 previously equilibrated with 1 M sodium acetate pH 5.0 at room temperature. The first protein fraction (P1G10) containing the bulk proteolytic activity was pooled and concentrated by ultrafiltration (10,000 Da pore size) and stored at −20 °C until used. The uniform composition of P1G10 has been demonstrated earlier by HPLC and SDS-PAGE electrophoresis [46] and the peptide isoforms comprising P1G10 were described [47]. The standard protein concentration and amidase activity of this fraction were 8.39 ± 0.39 mg/mL and 13.5 ± 0.5 nM/μg·×·min, respectively.

### 4.4. P1G10 Formulation

The vehicle used was a hydrosoluble base containing 5% propyleneglycol, 5% monoolein, and 0.25% hydroxyethylcellulose gel base (Natrosol) (pH 6.5). A 10% aqueous solution of P1G10 was dispersed into the hydrosoluble vehicle to a final concentration of 0.1% or 1.0 % *w*/*v*. The control sample contained a similar volume of saline dispersed directly in the vehicle. The formulations were stored at 4 °C.

### 4.5. Antioxidant Activity of P1G10

Free radical scavenging activity of P1G10 was determined using the 2,2-diphenyl-1-picrylhydrazyl (DPPH) assay [48]. Various dilutions of triplicate samples were added to 1 mL DPPH (0.5 mM) in aluminum foil wrapped test tubes. Absorbance was measured at 517 nm after 30 min incubation in the dark at room temperature and butylhydroxytoluene (BHT) was used as reference. The % ability to scavenge DPPH radical was calculated by the following equation:$$\mathrm{DPPH}\;\left(\%\right)=\;\left[\left(\frac{Abs_{control\;}-\;Abs_{sample}}{Abs_{control}}\right)\times 100\right]$$
where, *Abs_control_* is the absorbance of DPPH radical + methanol; *Abs_sample_* is the absorbance of DPPH radical + sample divided by the *Abs_control_*.

### 4.6. Exposure to Single UVB Dose

The source of UVB irradiation was a Cole-Parmer (USA) ultraviolet lamp with an emission peak at 312 nm (measured in Ocean Optics spectrometer and OOI Base32 Software). The lamp was mounted at 13 cm distance from a table top where mice were irradiated. The UVB output was determined using a calibrated CMP22 radiometer Kipp and Zonen. The irradiance was set at 8 W m^−2^ and the dose applied was 2.4 J cm^−2^ [49].

The animals were randomly divided into five groups (*n* = 5/group). Before irradiation, the animals were anesthetized with an i.p. injection containing ketamine (80 mg kg^−1^) and xylazine (10 mg kg^−1^). Thereafter, each animal was placed in polyvinyl chloride chamber (15 × 25 mm^2^) and received a dorsal single dose of 2.4 J cm^−2^ of UVB. The erythema score represents the redness intensity of the irradiated area [50]. Then, at different intervals (0, 6, 12, 24, and 48 h), the animals were anaesthetized, and skin fragments were excised and perpendicularly sectioned.

To evaluate the topic effect of P1G10 five groups were constituted: 1–control untreated; 2–control UV-B irradiated; groups 3, 4 and 5 were UVB-irradiated and treated with the vehicle, 0.1%, or 1.0% P1G10, respectively. The ointment was spread homogenously onto the injured skin from mice. Twenty-four h after irradiation and associated treatments, the erythema was scored. Then, the dorsal skin from each animal was excised for biochemical and histopathological analysis, as described above. The results are representative of three independent experiments.

### 4.7. Analysis of Oxidative Stress

Skin-tissue fragments were triturated in liquid N_2_ for 3 min and dissolved in potassium phosphate buffer (0.1 M, pH 6.5). The suspension was centrifuged for 10 min at 13,000× *g* at 4 °C. The protein content in supernatants was measured by Bradford [51].

### 4.8. Generation of Intracellular Reactive Oxygen Species (ROS)

ROS levels were detected using the oxidation sensitive dye 2,7-dichlorofluorescin diacetate (DCFH-DA, Sigma-Aldrich) assay, as previously described [47,52]. The fluorescence of DCF in PBS was detected at 485 nm using an emission wavelength of 535 nm in a fluorescence microplate reader (Synergy, Biotek Instrument). A standard curve with oxidized dichlorofluorescein (DCF) was created to express the results as µM mg^−1^ tissue.

#### 4.8.1. Superoxide Dismutase Activity (SOD)

The method [53], involving the generation of superoxide by pyrogallol autoxidation and inhibition of the superoxide dependent reduction of tetrazolium dye MTT [3-(4,5-dimethyl-thiazol-2-yl) 2,5-diphenyl tetrazolium bromide] to formazan, was measured at 570 nm. One SOD unit inhibits the increase in absorbance by 50% at 550 nm produced by a control sample without SOD under the assaying condition. Results are expressed in U mg^−1^ protein.

#### 4.8.2. Catalase Activity (CAT)

This measurement is based on the decomposition of H_2_O_2_ as described by Shangari and O’Brien [54]. Ten µL homogenate was added into a reaction mixture containing 7.5 mM H_2_O_2_ buffered in potassium phosphate (0.1 M, pH 7.4). The reduction in absorbance at 240 nm was scored after 3 min incubation at 37 °C and compared to a blank without the sample, and expressed according to the formula:$$\frac{\mathrm{Units}}{\mathrm{mL}}=\left[\left(\frac{\Delta A}{min_{blank}}-\frac{\Delta A}{min_{sample}}\right)\times d\times \frac{v}{\left(V\times \varepsilon \right)}\right]$$

The activity is expressed as Units mL^−1^ of catalase (sample volume), Δ*A* = change in absorbance, *d* = sample dilution, *v* = volume of reaction, *V* = sample volume in the reaction and *ε* = 2.35 × 10^5^ M^−1^ cm^−1^ (catalase molar extinction coefficient). The results referred to the protein content were expressed as percentage from control.

#### 4.8.3. Glutathione Peroxidase Activity (GSH-Px)

The reaction mixture consisting of 100 mM potassium phosphate buffer, 1.1 mM sodium azide, 2.1 mM EDTA, 0.15 M reduced glutathione, 8.4 mM NADPH, and 100 × UmL^–1^ glutathione reductase was incubated at 37 °C for 5 min with appropriate aliquots of tissue homogenates. The assay started with the addition of 2.2 mM H_2_O_2_. The absorbance of the samples was measured at 340 nm in a UV spectrophotometer (Shimadzu Corp. Kyoto, Japan). The data were registered at 15 s intervals during 300 s. GSH-Px activity was expressed as µM mg^−1^ tissue as described [55].

#### 4.8.4. Reduced Glutathione Content (GSH)

Samples were precipitated with 10% trichloroacetic acid, centrifuged, and the pellet discarded. Then, 1 mg mL^−1^ solution of *o*-phthalaldehyde dissolved in methanol was added to supernatants, previously diluted with 0.1 M sodium phosphate buffer, 5 mM EDTA, pH 8.0. After 15 min incubation at room temperature, the fluorescence was recorded in a plate reader (Synergy, Biotek Instrument, Winooski, VT, USA) at 350 nm excitation and 420 nm emission). GSH concentration was determined using a GSH standard curve and expressed as µM mg^−1^ tissue [56].

#### 4.8.5. Malondialdehyde Content (MDA)

Skin homogenates were mixed with 10 mM 2-thiobarbituric acid (EMD Millipore Corporation, Sao Paulo, Brazil) solution containing 75 mM K_2_HPO_4_, pH 3 (Sigma-Aldrich), 2.5 mM butylatedhydroxytoluene (BHT, Sigma-Aldrich) and incubated in a dry bath for 15 min, 100 °C. The reaction products were extracted with n-butanol and MDA was determined in the organic phase at 532 and 600 nm (Shimadzu, Japan). MDA concentration was determined using a standard curve (1.1.2-trimethoxypropane–TMB, Sigma-Aldrich) and expressed as µM mg^−1^ tissue [57].

### 4.9. Cytokines Analysis

Tissue was homogenized in PBS pH 7.4 containing 0.05% Tween-20 (Sigma-Aldrich) and centrifuged at 10,000× *g* for 30 min. The supernatant was used to determine levels of IL-1β, IL8, TGF-β, and IL10 by ELISA [58]. The assays were performed using kits from (DuoSet, R&D Immunoassay kit) according to the manufacturer’s instructions. The cytokines were expressed as pg·mg^−1^ tissue.

### 4.10. Myeloperoxidase Activity (MPO)

The sediment recovered after centrifugation from tissue homogenates (see cytokine assay above) was suspended with buffer for myeloperoxidase assay (MPO) to monitor neutrophil, as described by [58]. Tissue homogenized in buffer 0.02 M NaPO_4_, pH 4.7 was centrifuged at 12,000× *g* for 10 min. Pellets were suspended in 0.05 M NaPO_4_ buffer (pH 5.4) containing 0.5% hexadecyltrimethylammonium bromide followed by three freeze–thaw cycles using liquid N_2_. After addition of 1.6 mM tetramethylbenzidine and 0.3 mM H_2_O_2_ to supernatant samples, MPO activity was measured by the change in absorbance at 450 nm. Results are expressed as percentage of basal MPO activity.

### 4.11. Gelatin Zymography

Gelatinolytic activity of matrix metalloproteinase-9 (MMP-9) was assayed by zymography, as previously described by [59]. An aliquot of sample extract was homogenized in 0.1 M potassium phosphate buffer, pH 6.5 and incubated with electrophoresis sample buffer for 10 min at room temperature. Samples were electrophoresed in 10% SDS-PAGE slab gels containing 0.1% gelatin, at 90 V during 90 min. Removal of SDS from gel was done by washing with 2.5% Triton X-100 for 1 h at room temperature followed by overnight incubation in buffer (50 mM Tris–HCl, pH 7.6, containing 10 mM CaCl_2_ and 0.15 M NaCl). Digital images and densitometric analyses of gels were obtained with ImageQuant LAS4000 (GE-Healthcare), and the MMPs activity was quantified with ImageQuant software. The results are expressed as percent inhibition of the gelatinolytic activity induced by the extracts using as reference saline treated samples. Molecular mass standards (BioRad Laboratories) were electrophoresed in parallel with samples to assess protein size.

### 4.12. Western Blot

Variations in pAkt, Akt, GAPDH, COX2 and p53 were evaluated by Western blot using anti-pAkt, anti-Akt, and anti-GAPDH (Cell Signaling Technology), anti-COX_2_, and anti-p53 (Abcam). Skin samples were mechanically triturated and powdered in a mortar using liquid N_2_ for 3 min. The protein was extracted in 500 µL ice-cold lysis buffer (20 mM Tris–HC1, pH 7.5, 150 mM NaCl, 1 mM EDTA, 1 mM EGTA, 1% Triton X-100, 2.5 mM sodium pyrophosphate, 1 mM b-glycerophosphate, 1 mM Na_3_VO_4_, 1 mg/L leupeptin, 1 mM PMSF—Sigma-Aldrich). After centrifugation at 10,000× *g* at 4 °C for 30 min, the protein concentration was assessed by Bradford assay [46,51]. The proteins (30 µg) were resolved on a 10% SDS–PAGE and electroblotted onto PVDF membranes. The membranes were then blocked with 5% BSA in TBST at room temperature for 2 h and incubated with rabbit or mouse antibodies against p-Akt (1:1000), Akt (1:2000), p53 (1:1000), COX-2 (1:500), and GAPDH (1:10.000) overnight at 4 °C. The secondary anti rabbit antibodies were diluted 1:10,000 in TBST and incubated with membranes for 2 h at 25 °C. The membrane was processed for chemiluminescent detection using Luminata Forte Western HRP Substrate (Invitrogen, California, EUA) according to the manufacturer’s instructions and the reactive bands were detected in ImageQuant LAS 4000 (G&E Healthcare).

### 4.13. Histological Analysis

Tissue sections were processed for microscopy and stained with hematoxylin and eosin (H&E) [60] for morphometric evaluation or Gomori trichrome [61] staining, for analysis of collagen fiber under a light microscope. All histological sections were analyzed with 10x and 40x objectives and between 5 to 30 images of sections were captured by a Q-Color3 digital camera (Olympus). Algorithms from the KS300 software (Carl Zeiss, Oberkochen, Germany) were used for image processing, as described by [62]. Cell number was determined by scoring nuclei in all cell types in the hypodermis (1.6 × 10^6^ mm^−2^, area). Skin thickness analysis was made with a 10× objective in five different random lesions and their average calculated. In Gomori trichrome stained slices, blue color intensity represents the density of collagen at 40× magnification. Mean density of each group was defined as the collagen density of the irradiated area versus the collagen density of equal area in normal tissue.

The streptavidin-peroxidase (SP) method was used for immunohistochemical staining using the Kit Ultra vision large volume anti-polyvalent detection system, HRP-Lab Vision, DAKO, as described [63]. The antibody used was Anti-caspase-3 (dilution 1:600—Cell Signaling Technology, Danvers, MA, USA). The level of caspase-3 was quantitated in epidermis cells.

### 4.14. Statistical Analysis

The data from triplicate experiments are expressed as the mean ± standard error (SEM). Each experiment was repeated three times. Differences between three or more groups were analyzed using one-way ANOVA, followed by Newman–Keuls multiple comparisons test. Statistical significance was set at * *p* < 0.05; ** *p* < 0.01 or *** *p* < 0.001.

## 5. Conclusions

The proteolytic fraction (P1G10) from *V. cundinamarcensis* exerts topic antioxidant and anti-inflammatory effects on mice dermal lesions induced by UVB radiation. These effects are mediated by increase in scavenging agents and/or the enzymes involved in their synthesis, and a reduction of pro-inflammatory agents. The changes induced by P1G10 provided a protection minimizing the symptoms triggered by UV radiation.

## Figures and Tables

**Figure 1 ijms-20-04373-f001:**
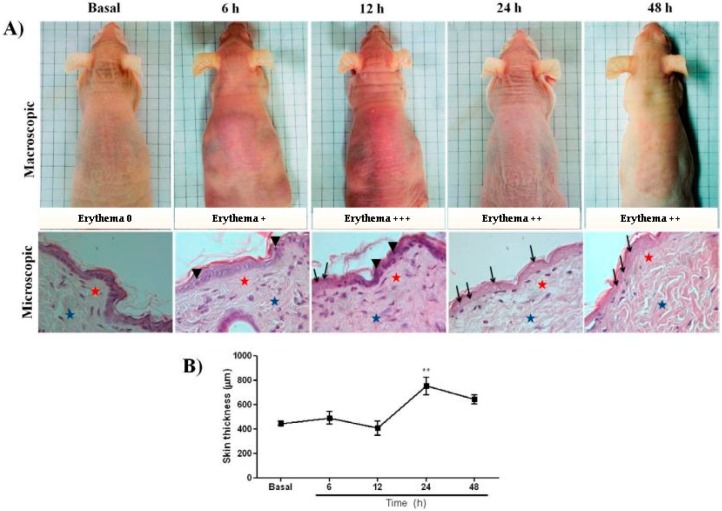
(**A**) Macro- and microscopic changes induced by UVB in skin of *hairless* mouse. Dorsal skin was exposed to a single UVB dose (2.4 J cm^−1^) and changes were monitored at various intervals. Skin biopsies were kept in 10% buffered formalin and processed for histopathological examination. Representative photomicrographs using 40× objective of Haematoxylin-eosin at 6, 12, 24, and 48 h post UV were compared with basal group. Apoptotic cell = arrowhead; enucleated cells = arrows; superficial dermis = red stars; deep dermis = blue stars). (**B**) Morphometric analysis of basal, 6, 12, 24, and 48 h after irradiation. The data represent total skin thickness (10× objective): mean ± SEM of 6–8 animals in each group (** *p* < 0.01 difference versus basal, ANOVA, Newman–Keuls post-test).

**Figure 2 ijms-20-04373-f002:**
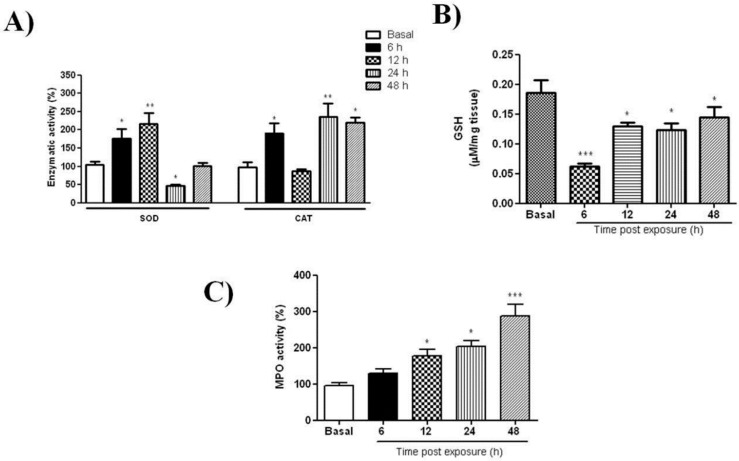
Antioxidant mediators following skin UVB-radiation. Dorsal skin was exposed to a single dose of UVB (2.4 J cm^−1^) and changes were analyzed at different intervals. Activity profiles of: (**A**) superoxide dismutase (SOD) and catalase activity (CAT); (**B**) glutathione content (GSH) level; (**C**) myeloperoxidase activity (MPO). For details see Materials and Methods. Data are expressed as mean ± SEM of 6–8 animals per group (* *p* < 0.05, ** *p* < 0.01 and *** *p* < 0.001 difference versus Basal, ANOVA, Newman–Keuls post-test).

**Figure 3 ijms-20-04373-f003:**
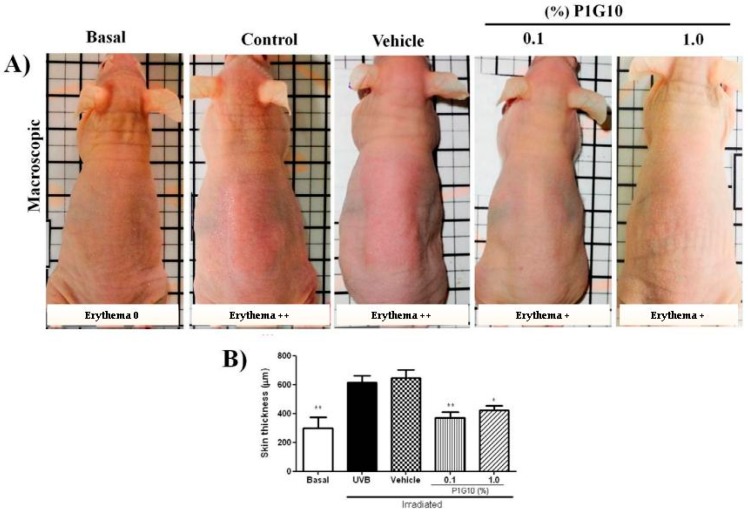
P1G10 inhibits erythema induced by UVB. Mice were exposed to a single UVB dose (2.4 J cm^−1^) followed by topical application of P1G10. (**A**) Erythema intensity of basal, control, Natrosol gel vehicle, 0.1% and 1.0% P1G10 dispersed in Natrosol. Skin biopsies were fixed and stored in 10% buffered formalin and processed for histological and morphometric analysis. (**B**) Epidermal thick analysis of: Basal, control, vehicle, 0.1% and 1.0% P1G10 24 h after irradiation. For details see Materials and Methods. The data represent total skin thickness (10×objective): mean ± SEM of 6–8 animals per group (** *p* < 0.01 and * *p* < 0.05 difference versus vehicle, ANOVA, Newman–Keuls post-test).

**Figure 4 ijms-20-04373-f004:**
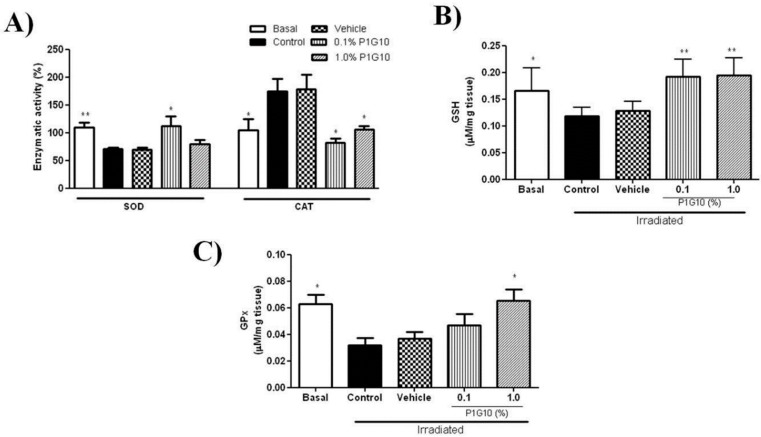
Effect of P1G10 on the activity of SOD, CAT, MPO, and GSH in UVB-irradiated skin. Mice were exposed to a single UVB dose (2.4 J cm^−1^) followed by treatment with Natrosol gel (vehicle) or P1G10 (0.1% and 1.0%). Skin of euthanized mice was collected 24 h after irradiation and processed for analysis (**A**) SOD, CAT, (**B**) GSH and (**C**) glutathione peroxidase (GSH-Px). For details see Materials and Methods. Results represent the mean ± SEM of 6–8 animals per group (* *p* < 0.05 and ** *p* < 0.001 difference versus vehicle, ANOVA, Newman–Keuls post-test).

**Figure 5 ijms-20-04373-f005:**
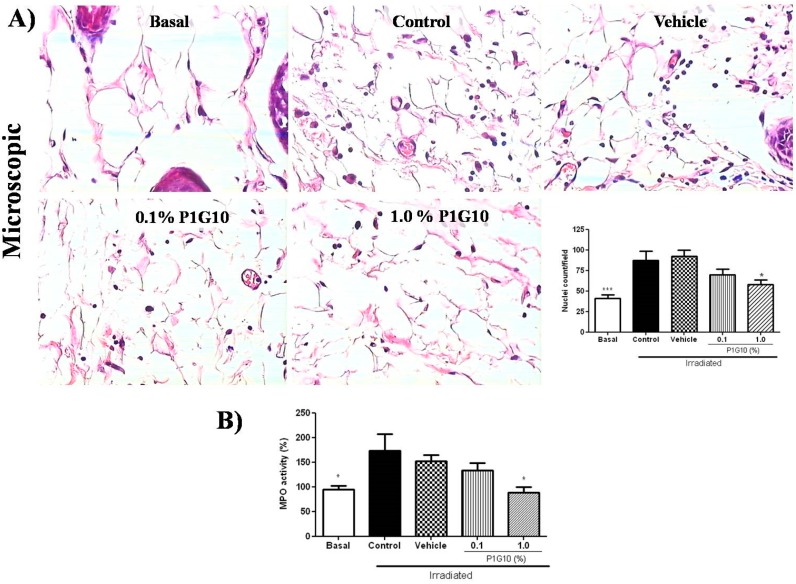
The effect of P1G10 on the infiltration of inflammatory cells in UVB-irradiated mice. Biopsies were preserved in 10% buffered formalin for histological and morphometric analysis. (**A**) Photomicrographs illustrate the nuclei density of hypodermis in slides stained with H&E (40x objective): Basal, control, vehicle, 0.1% and 1.0% P1G10. The data quantifies hypodermal nuclei from the image. (**B**) Infiltrating neutrophils were quantified by myeloperoxidase assay (MPO). Results represent the mean ± SEM of 6–8 animals per group (* *p* < 0.05 and *** *p* < 0.001 difference versus vehicle, ANOVA, Newman–Keuls post-test).

**Figure 6 ijms-20-04373-f006:**
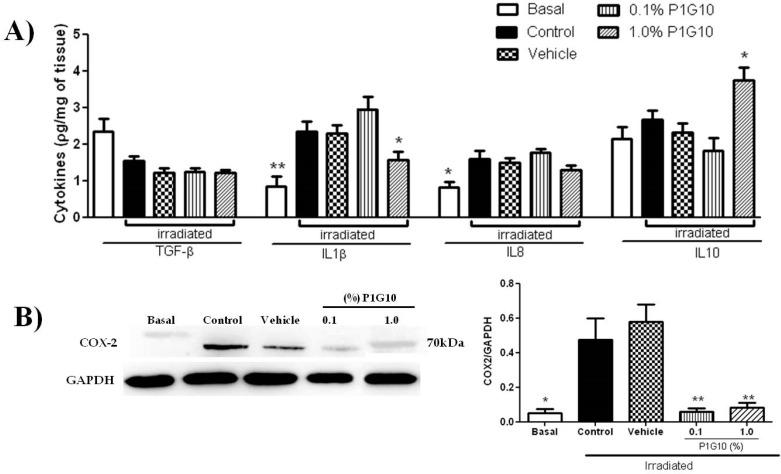
The effect of P1G10 on cytokines and COX2 activity in irradiated mice. Mice were exposed to UVB radiation (2.4 J cm^−1^) and treated with Natrosol gel, (vehicle) or P1G10. After 24 h, skin samples were dissected and processed for cytokines analysis (**A**) quantitative ELISA of TGF-β, IL1β, IL8, and IL10. (**B**) COX_2_ activity was determined in cell lysates using Western blot analysis. COX_2_ level is expressed relative to GAPDH using ImageQuant software. Results represent the mean ± SEM of 6–8 animals per group (* *p* < 0.05 and ** *p* <0.01 difference versus vehicle, ANOVA, Newman–Keuls post-test).

**Figure 7 ijms-20-04373-f007:**
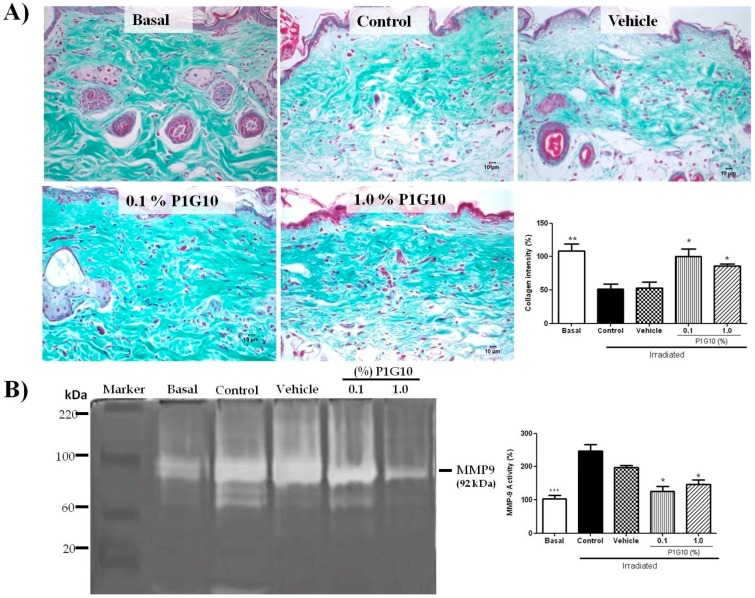
Changes in collagen and MMP-9 activity induced by P1G10 after UVB-irradiation. Mice were exposed to UVB (2.4 J cm^−1^) and treated with Natrosol gel (vehicle) or 0.1% and 1% P1G10 for 24 h post-irradiation. Biopsies were preserved in 10% buffered formalin for histological analysis. (**A**) Photomicrographs illustrate collagen fibers from dermis in slides stained with Gomori trichrome (40× objective). The graph represents the area corresponding to collagen fibers obtained from the image after digital processing. (**B**) MMP-9 activity was assessed by zymography and data expressed as percentage MMP-9 activity relative to basal value (100%). Results represent the mean ± SEM of 6–8 animals per group (* *p* < 0.05, ** *p* < 0.01 and *** *p* < 0.001 difference versus vehicle, ANOVA, Newman–Keuls post-test).

**Figure 8 ijms-20-04373-f008:**
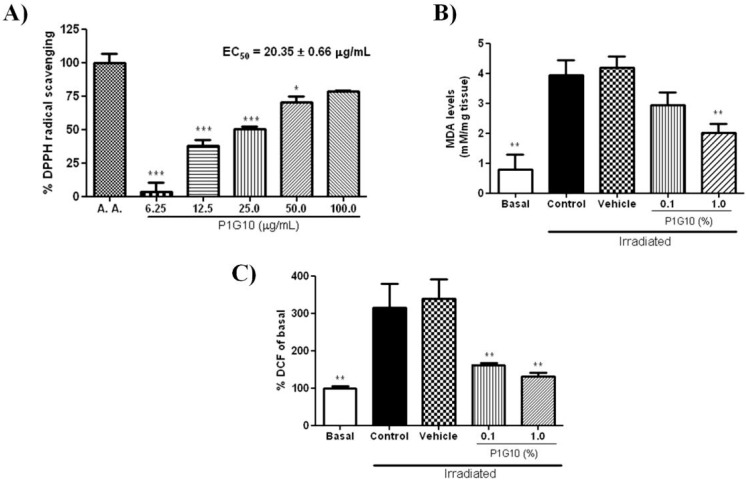
The effect of P1G10 on mediators of oxidative stress in UVB-irradiated skin. Mice were exposed to UVB (2.4 J cm^−1^) and treated with Natrosol gel (vehicle) or 0.1% and 1% P1G10. After 24 h, the skin of euthanized animals was dissected and processed for analysis. (**A**) Antioxidant activity measured by DPPH assay. (**B**) Content of malondialdehyde (MDA); (**C**) production of dichlorofluorescein (DCF). For details see the Materials and Methods. Data are expressed as mean ± SEM of 6–8 animals per group (* *p* < 0.05, ** *p* < 0.01, *** *p* < 0.001 differences versus Vehicle, ANOVA, Newman–Keuls post-test).

**Figure 9 ijms-20-04373-f009:**
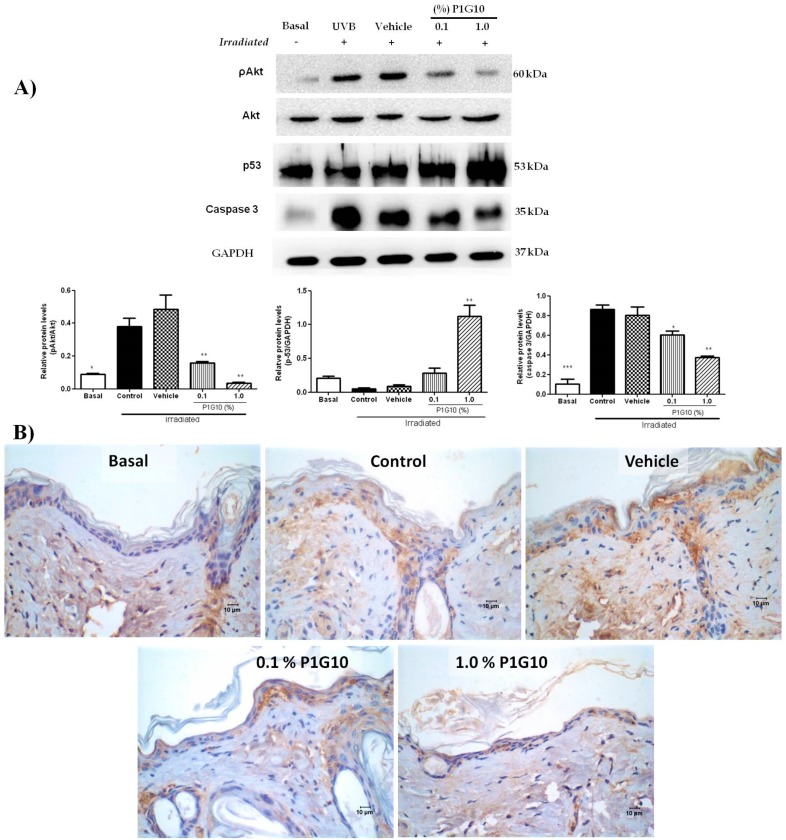
P1G10 alters the expression of pakt, p53, and caspase-3 in irradiated skin. Mice were exposed to UVB (2.4 J cm^−1^) and treated with Natrosol gel (vehicle) or 0.1% and 1% P1G10. After 24 h, the skin of euthanized animals was dissected and processed for analysis. Cell lysates were used to assess pakt, p53, and caspase-3 (**A**). The data show a western blot and densitometric analysis. Protein levels were normalized to GAPDH using ImageQuant software. (**B**) A brown color indicates the presence of caspase-3 positive cells in irradiated samples. Images captured with 40× objective, Scale = 10 µm. Results represent the mean ± SEM of 6–8 animals per group (* *p* < 0.05, ** *p* < 0.01 and *** *p* < 0.001 difference versus vehicle, ANOVA, Newman–Keuls post-test).

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
