# Peer review of "P1G10, the Proteolytic Fraction from Vasconcellea cundinamarcensis, Stimulates Tissue Repair after Acute Exposure to Ultraviolet B Radiation"

_ijms, 2019, doi:10.3390/ijms20184373_

Round 1
Reviewer 1 Report
The paper “P1G10 the Proteolytic Fraction from Vasconcellea cundinamarcensis Stimulates Tissue Repair After Acute Exposure to Ultraviolet B Radiation” by Freitas et al. is very interesting as it is focused on the use of natural substances for the treatment or to prevent the skin damages induced by UV radiations.
The use of natural substances has attracted a great attention in the last few decades especially for the treatment of skin disorders connected to inflammation and oxidative stress.
The authors speaking about this aspect should also present other approaches performed by other researchers. The use of nanocarriers for example incorporating natural substances has been widely used and presented in scientific papers, so I suggest adding also some references. as an example:
“Combination of argan oil and phospholipids for the development of an effective liposome-like formulation able to improve skin hydration and allantoin dermal delivery” by Manca et al. International Journal of Pharmaceutics, 2016
“Inhibition of skin inflammation by baicalin ultradeformable vesicles” by Mir-Palomo et al. International Journal of Pharmaceutics, 2016
“Nanodesign of new self-assembling core-shell gellan-transfersomes loading baicalin and in vivo evaluation of repair response in skin” By Manconi et al. Nanomedicine: Nanotechnology, Biology, and Medicine 2018.
In general, the paper is well written, and can be added in the present form, I just suggest a careful check for the English language.
Author Response
Reviewer 1) The reviewer requests information on possible nano-formulations containing the active principle A statement is included in the revised version mentioning the possible production of nanoparticles containing P1G10 to ameliorate the release of the active principle. “The possibility of modulating the anti-inflammatory action by incorporating an active agent into liposomes is being established [43, 44] and there are reports [45] describing the production of phospholipid vesicles containing bromelain or papain without impairing the proteolytic activity, thus we envision future studies to produce nanoparticles containing P1G10 for treatment of the inflammatory symptom.”
43. Mir-Palomo, S.; Nácher, A.; Díez-Sales, O.; Vila-Busó, O.; Caddeo, C.; Manca, M.L.; Manconi, M.; Fadda, A.M.; Saurí, A.R.; Inhibition of skin inflammation by baicalin ultradeformable vesicles. Int. J. Pharm. 2016, 511, 23– 29
44. Manconi, M.; Manca, M.L.; Caddeo, C.; Valenti, D.; Cencetti, C.; Diez-Sales, O.; Nacher, A.; Mir-Palomo, S.; Terencio, M.C.; Demurtas, D.; Gomez-Fernandez, J.C.; Aranda, F.J.; Fadda, A.M.; Matricardi, P. Nanodesign of new self-assembling core-shell gellan-transfersomes loading baicalin and in vivo evaluation of repair response in skin. Nanomedicine. 2018,14:569–579.
45. Sahu, K.; Kaurav, M.; Pandey, R.S. Protease loaded permeation enhancer liposomes for treatment of skin fibrosis arisen from second degree burn. Biomed. Pharmacother. 2017, 94:747–757
Reviewer 2 Report
P1G10 had been published at various journal which present Antifungal activity and as enhances wound healing of diabetic foot ulcers. P1G10 is amazing mixture, it would be worthy investigate.
Previous studies demonstrated that P1G10 present Antifungal activity and as enhances wound healing of diabetic foot ulcers. Authors may discuss the other functions that different from yours finding.
Moreover, P1G10 is a proteolytic fraction from Vasconcellea cundinamarcensis latex, authors just refer reference 43, if discuss more about the identification components, it would be more interesting.
Other the other hand, the resolution of figures should be improving.
The Proteolytic Fraction from Latex of Vasconcellea cundinamarcensis (P1G10) Enhances Wound Healing of Diabetic Foot Ulcers: A Double-Blind Randomized Pilot Study. Adv Ther. 2018 Apr;35(4):494-502.
Antifungal activity of proteolytic fraction (P1G10) from (Vasconcellea cundinamarcensis) latex inhibit cell growth and cell wall integrity in Botrytis cinerea. Int J Food Microbiol. 2019 Jan 16;289:7-16.
Author Response
Reviewer 2 requests additional information on the biochemical properties of P1G10, thus; the revised version includes a reference with the characterization of P1G10 components, as follows;
“The uniform composition of P1G10 has been demonstrated earlier by HPLC and SDS-PAGE electrophoresis [46] and the peptide isoforms comprising P1G10 is being described [47].”
Teixeira R.D; Ribeiro H.A; Gomes M.T; Lopes M.T; Salas C.E. The proteolytic activities in latex from Carica candamarcensis. Plant Physiol. Biochem. 2008, 46, 956-61.